# Dual Inoculation with *Rhizophagus irregularis* and *Bacillus megaterium* Improves Maize Tolerance to Combined Drought and High Temperature Stress by Enhancing Root Hydraulics, Photosynthesis and Hormonal Responses

**DOI:** 10.3390/ijms24065193

**Published:** 2023-03-08

**Authors:** Antonia Romero-Munar, Ricardo Aroca, Angel María Zamarreño, José María García-Mina, Noelia Perez-Hernández, Juan Manuel Ruiz-Lozano

**Affiliations:** 1Departamento de Microbiología del Suelo y Sistemas Simbióticos, Estación Experimental del Zaidín (CSIC), Profesor Albareda N° 1, 18008 Granada, Spain; 2Departmento de Biología Ambiental, Facultad de Ciencias, Universidad de Navarra, Irunlarrea No 1, 31008 Pamplona, Spain; 3Estación Biologia de Doñana (CSIC), Avd. Americo Vespucio 26, 41092 Seville, Spain

**Keywords:** aquaporin, arbuscular mycorrhiza, combined drought and heat stress, maize, PGPR, root hydraulic conductivity

## Abstract

Climate change is leading to combined drought and high temperature stress in many areas, drastically reducing crop production, especially for high-water-consuming crops such as maize. This study aimed to determine how the co-inoculation of an arbuscular mycorrhizal (AM) fungus (*Rhizophagus irregularis*) and the PGPR *Bacillus megaterium* (Bm) alters the radial water movement and physiology in maize plants in order to cope with combined drought and high temperature stress. Thus, maize plants were kept uninoculated or inoculated with *R. irregularis* (AM), with *B. megaterium* (Bm) or with both microorganisms (AM + Bm) and subjected or not to combined drought and high temperature stress (D + T). We measured plant physiological responses, root hydraulic parameters, aquaporin gene expression and protein abundances and sap hormonal content. The results showed that dual AM + Bm inoculation was more effective against combined D + T stress than single inoculation. This was related to a synergistic enhancement of efficiency of the phytosystem II, stomatal conductance and photosynthetic activity. Moreover, dually inoculated plants maintained higher root hydraulic conductivity, which was related to regulation of the aquaporins *ZmPIP1;3*, *ZmTIP1.1*, *ZmPIP2;2* and *GintAQPF1* and levels of plant sap hormones. This study demonstrates the usefulness of combining beneficial soil microorganisms to improve crop productivity under the current climate-change scenario.

## 1. Introduction

The global world population is predicted to rise to between 9.7 billion and 10 billion by 2050 [1]. Considering that the rising human population is occurring under a climate-change scenario, it is thus highly important to increase crop tolerance to the new adverse environmental conditions in order to secure food production [1,2]. Indeed, both global warming and reduced rainfall cause a negative impact on plant growth, development and productivity [3]. In Mediterranean regions, drought usually occurs concomitantly with high temperatures. Indeed, high temperature has been proposed as one of the main environmental factors that hinders plant growth, produces yield reduction, causes decreases in fertile lands and prevents optimal water use [4,5]. The stress produced by high temperatures damages plant metabolism and development and is becoming highly relevant, producing the loss of cultivable areas on Earth, with important social and economic effects [6,7]. In addition, the concurrence of drought with high temperature is more deleterious than drought alone and may lead to greater agricultural damage [8].

In this context, an important aspect that must be considered is that the plant response to a combined drought and heat stress is complicated by its prioritization for the more serious stress for the plant. For instance, stomata are closed prematurely by plants under drought conditions in order to prevent water loss, while, under heat stress, plants increase stomatal conductance to cool down the temperature of leaves by means of transpiration [8,9,10]. Thus, drought and heat stress have an adverse impact on plants, and their combined effect is higher than when taken individually [4]. Hence, understanding the mechanisms allowing higher crop productivity under combined water and heat stressed conditions is of main interest in order to guarantee food production in coming years [2,11].

To increase crop tolerance to adverse growing conditions, the use of natural resources is an outstanding approach in the context of the modern sustainable agriculture. This includes the use of beneficial soil microorganisms known to improve both plant development and productivity under limiting environmental conditions. Among these beneficial soil microorganisms, we would like to highlight the so-called arbuscular mycorrhizal fungi. These soil fungi establish a mutualistic symbiosis with the roots of most terrestrial plants, and this is called arbuscular mycorrhiza (AM). The AM symbiosis stimulates the plant physiology through a series of changes at the morphological and molecular plant levels. As a consequence, the host plant increases its ability to maintain vigor and survive under adverse conditions, being a successful example of a sustainable agricultural approach [3]. Other microorganisms that help plants to tolerate abiotic stresses are the so-called plant-growth-promoting rhizobacteria (PGPR). These bacteria may live in the rhizoplane or inside roots and have several mechanisms to promote plant growth and increase their tolerance to abiotic stresses, such as the reduction of ethylene levels; nutrient solubilization; production of some hormones, degradative enzymes and siderophores; or nitrogen fixation, among others [8,12]. In this context, the combined use of AM fungi and PGPRs depicts a highly sustainable strategy to enhance plant tolerance to adverse environmental conditions, such as heat stress and water limitation [13,14].

In most cases studied, the association between an AM fungi and a plant makes the host plant more tolerant to water limitation [15], and this has been attributed to an improvement of soil water-retention properties and soil structure, the uptake of water through the fungal hyphae and transfer to the host plant, protection against the oxidative damage generated by drought, a better osmotic adjustment in AM plants, the enhancement of plant gas exchange and water-use efficiency and a more efficient uptake of nutrients, as reviewed by [3,14,16,17]. In addition, the establishment of the AM symbiosis produces morphological changes in the roots of the host plant, involving cellular membranes and altering membrane-associated proteins such as aquaporins. These proteins are small channels located at different cell membranes that allow the passive crossing of small neutral molecules and water. Aquaporins constitute a large family in vascular plants subdivided in the following subfamilies: PIPs (plasma membrane intrinsic proteins), TIPs (tonoplast intrinsic proteins), NIPs (nodulin 26-like intrinsic proteins) and SIPs (small basic intrinsic proteins) [18,19]. Some plants also contain the uncharacterized XIPs (X intrinsic proteins) [20]. Aquaporins constitute the main pathway for water passage through cell membranes [19,21] and allow a rapid regulation of membrane water permeability. This influences root hydraulic conductivity and the whole-plant water balance during episodes of water deficit [22,23]. Moreover, the high interest of aquaporins for plant physiology comes from the fact that, besides water, certain aquaporins allow the membrane movement of other small molecules with physiological importance, such as urea, ammonia, H_2_O_2_, CO_2_, metalloids, oxygen or some ions [21,24,25]. Moreover, it has been emphasized that, in positive plant–microbe interactions involving rhizobia, AM fungi and PGPR, aquaporins play important roles in nitrogen fixation, nutrient transport, improving water status and increasing abiotic stress tolerance [26].

The results from previous studies have shown that the AM symbiosis has the capacity of altering root hydraulic conductivity (Lpr), enhancing it mostly under stress conditions [27,28,29,30,31,32,33,34,35], with involvement of plant aquaporins in these processes [14]. Recently, it has been shown that the presence of the AM fungus in the root increases the water permeability of root cells, related to the induction of some aquaporin genes and increase of the phosphorylation status of PIP2s, which implies a higher activity of their water channels [34]. Moreover, the presence of a mycorrhizal fungus significantly modified the radial transport of water within the root system [33]. Regarding the improvement of root hydraulic properties by PGPR under osmotic stress conditions, our research group found a *Bacillus megaterium* strain that was able to improve Lpr under salinity stress in maize plants by also increasing the amount of PIP proteins in the roots [36]. Other studies have shown that the effects of the AM symbiosis on root Lpr are altered by co-inoculation with a PGPR, enhancing or decreasing it depending on the genetic characteristics of the tomato line [37]. However, very few studies have investigated the possible involvement of aquaporins on the PGPR enhance osmotic stress tolerance in plants [26]. In addition, we found that the positive effect of a *B. megaterium* strain on tomato-plant growth was dependent on the plant ABA content and ethylene sensitivity. In fact, this particular strain was unable to promote plant growth in ABA-deficient tomato plants [38] or in ethylene-insensitive tomato plants [39]. Moreover, as the plant responses to stress are regulated by a hormonal crosstalk [40,41], some of the effects of AM fungi and PGPR on plant performance under drought stress (including root hydraulic properties) have been related to changes in the plant hormonal content [30,42,43,44]. It is known that abscisic acid (ABA) alters transpiration and root hydraulic conductance [45]. Jasmonates are involved in plant development and also have a role against biotic and abiotic stresses [30,46]. Auxins are implicated in the process of AM fungal colonization [43,47] and development of arbuscules [48], as well as in plant responses to drought [35,49]. Salicylic acid (SA) is involved in nitrogen metabolism, regulation of photosynthesis, antioxidant defense system and plant–water relations [32,50,51].

Regarding the effects of high temperature on root water transport, little information is available. However, it seems that there is a beneficial short-term effect due to an enhanced apoplastic mass flow and a negative long-term effect as a result of deterioration of root physiological functions [52]. No information about the role of AM fungi and/or PGPR on that was found.

Therefore, the global objective of this study was to determine how the co-inoculation of an AM fungus and a PGPR alters the radial water movement and physiology in the host plant in order to cope with combined drought and high temperature stress, as well as the role of plant aquaporins and phytohormones in these processes. Our starting hypothesis was that the co-inoculation of AM fungi and PGPR can act on the host plant in a concerted manner to alter the plant water relations and its physiology in order to cope better with the combined stressful conditions.

## 2. Results

### 2.1. Plant Growth and AM Root Colonization

Plants from both AM treatments (inoculated only with the AM fungus or in combination with *B. megaterium*) grew more than control uninoculated plants (Figure 1A). This was significant both under unstressed conditions (10% of increase) and also under combined drought + temperature (D + T) stress (19% of increase). Under both conditions, the highest shoot dry weight was achieved in plants dually inoculated with the AM fungus plus *B. megaterium* (AM + Bm), although no significant differences from plants singly inoculated with the AM fungus were observed. The single inoculation with *B. megaterium* did not improve plant growth as compared to uninoculated controls both under unstressed and combined D + T stress.

The combined D + T stress significantly decreased the shoot dry weight in all treatments, but the decrease was lower in AM treatment (9.5%) or in AM + Bm treatment (7.7%) than in uninoculated control plants (16.0%) or in plants inoculated only with Bm (16.6%).

Root dry weight was significantly reduced by the combined D + T stress in non-AM treatments (especially in uninoculated controls), while the decrease was not significant in both AM treatments (Figure 1B). Thus, the shoot-to-root ratio was maximum in plants dually inoculated with AM + Bm subjected to combined D + T stress and was enhanced significantly by the stresses applied only in uninoculated control plants.

The AM root colonization level was statistically similar in all the inoculated treatments, ranging from 70% of mycorrhizal root length in unstressed plants dually inoculated with AM + Bm to 76% of mycorrhizal root length in unstressed plants singly inoculated with the AM fungus. The combined D + T stress did not affect this parameter. No AM root colonization was observed in the uninoculated control plants or in those singly inoculated with Bm.

### 2.2. Shoot Water Content

Under optimal conditions, the highest shoot water content (86.5%) was found in AM plants, followed by plants dually inoculated (AM + Bm) or plants singly inoculated with Bm (Figure 2A). The lowest value (84.0%) was found in uninoculated control plants. Drought stress significantly decreased this parameter in all treatments, and, again, the lowest value (82%) was found in uninoculated control plants.

### 2.3. Membrane Electrolyte Leakage

Under optimal growth conditions, the membrane electrolyte leakage was similar in all treatments (Figure 2B). When plants were subjected to combined D + T stress, this parameter was highly enhanced in uninoculated plants (by 113%) and in those inoculated only with Bm (by 102%). The increase was lower in plants singly inoculated with the AM fungus (by 76%) and even lower in those dually inoculated with AM + Bm (by 42%). Thus, under combined D + T stress, uninoculated control plants exhibited 67% more membrane electrolyte leakage than those dually inoculated with AM + Bm. It is noteworthy that dual inoculation with AM + Bm reduced the membrane electrolyte leakage more than single inoculation with Bm or single inoculation with the AM fungus, showing a cooperative effect of both microorganisms on this parameter.

### 2.4. Stomatal Conductance and Efficiency of Photosystem II

Stomatal conductance (gs) was affected significantly by the combined D + T stress, the microbial inoculation and their interactions (Figure 3A). Thus, the combined D + T stress significantly reduced this parameter in all treatments, but the reduction was low in both AM treatments (AM alone or AM + Bm), which decreased gs by 40% as compared to the corresponding plants cultivated under optimal conditions. In contrast, in uninoculated controls or plants singly inoculated with Bm, the reduction of gs values due to the stress applied was by 89% and 74%, respectively.

To quantify efficiency of photosystem II, we measured the light-adapted maximum quantum yield of PSII primary photochemistry (Fv′/Fm′). Under optimal conditions, few differences among treatments were observed for this parameter, although plants dually inoculated with AM + Bm exhibited a slightly higher value than uninoculated control plants (Figure 3B). However, when plants were subjected to combined D + T stress, this value showed important differences between both AM treatments and both non-AM treatments. Thus, the lowest value was found in uninoculated control plants (which decreased this parameter by 62% as compared to unstressed conditions). In contrast, both AM treatments maintained high values for this parameter and only decreased it by around 15%. Plants singly inoculated with Bm also had an important decrease in this parameter (by 56%) due to the combined D + T stress.

### 2.5. Photosynthetic Activity and Water Use Efficiency

The net photosynthesis (An) was significantly affected by the combined D + T stress, the microbial inoculation and their interactions (Figure 4A). Thus, combined D + T stress significantly reduced the An in all treatments, except in plants dually inoculated with AM + Bm, as these maintained An levels similar to unstressed plants. Plants singly inoculated with the AM fungus decreased this parameter by 30% as compared to the same plants cultivated under optimal conditions. In contrast, the reduction due to combined D + T stress was considerably higher in uninoculated controls (by 84%) and in plants singly inoculated with Bm (by 67%).

Regarding instantaneous water-use efficiency (iWUE), only the combined D + T treatment affected this parameter, increasing it in all microbial treatments as compared to plants cultivated under optimal conditions (Figure 4B).

### 2.6. Osmotic Root Hydraulic Conductivity (Lo)

The values of osmotic root hydraulic conductivity (Lo) showed important differences among treatments (Figure 5A). Thus, under optimal conditions, both AM treatments (fungus alone or in combination with Bm) showed higher values (by 56 and 33%, respectively) than plants singly inoculated with Bm and considerably higher (by 335 and 272%, respectively) than uninoculated control plants.

When plants were subjected to combined D + T stress the differences were even higher. Thus, AM + Bm plants enhanced Lo by 38% over single AM plants, by 227% over single Bm plants and by 1490% over uninoculated control plants. Single AM inoculation and single Bm inoculation also enhanced Lo over uninoculated control plants by 1053% and by 386%, respectively. It is noteworthy that plants dually inoculated with AM + Bm had significantly higher Lo values under combined D + T stress than under optimal conditions, emphasizing the important synergistic effect of both microorganisms on this parameter under stressful conditions.

### 2.7. Hydrostatic Root Hydraulic Conductivity (Lpr)

The effects of the imposed stresses or microbial inoculants on Lpr were less important than on Lo. In any case, under optimal conditions, both AM treatments (fungus alone or in combination with Bm) showed higher values (by 19 and 34%, respectively) than uninoculated control plants (Figure 5B). Single inoculation with B. megaterium did not significantly affect this parameter as compared to uninoculated control plants or to AM plants. When plants were subjected to combined D + T stress, again, the highest Lpr value was achieved in plants dually inoculated with AM + Bm. Single AM inoculation also improved this parameter (by 18%) over uninoculated control plants, while single Bm inoculation did not significantly affect this parameter as compared to uninoculated control plants or to AM plants.

It is noteworthy that plants dually inoculated with AM + Bm again exhibited similar high Lpr values under combined D + T stress than under optimal conditions.

### 2.8. Expression of Plant and Fungal Aquaporins

Some of the analyzed aquaporins (*ZmPIP1;1*, *ZmPIP2;4*, *ZmTIP2;3*, *ZmTIP4;1* and *ZmNIP2;1*) did not show significant alteration in gene expression due to the treatments applied in this study (inoculation with AM and/or Bm and application or not of a combined drought and high temperature stress). Thus, these data are not shown.

Among the other aquaporins analyzed, the most significant results were found for *ZmPIP1;3* and *ZmPIP2;2* (Figure 6A,B). Thus, under optimal conditions, the expression of *ZmPIP1:3* was higher in AM plants (inoculated only with the AM fungus or in combination with Bm). When plants were subjected to combined D + T stress, the expression of this gene increased significantly in plants singly inoculated with Bm and also in those dually inoculated with AM + Bm. This increase was significant as compared to the uninoculated control plants and also as compared to the same treatment under optimal conditions. Plants singly inoculated with the AM fungus enhanced the expression of this gene due to the stress only as compared to uninoculated control plants but not with the same treatment under optimal conditions. Under optimal conditions, the increase in the expression of this gene in AM plants was also significant as compared to uninoculated controls (Figure 6A).

The expression of *ZmPIP2;2* was inhibited under optimal conditions by the inoculation with Bm and even more in plants dually inoculated with AM + Bm (Figure 6B). In contrast, the inoculation of the AM fungus alone did not alter the expression of this gene under optimal conditions. When the plants were subjected to combined D + T stress, all the treatments had similar levels of gene expression. However, the combined D + T stress applied significantly inhibited the expression of this gene in all treatments except in plants dually inoculated with AM + Bm, which already had the lowest expression levels.

Regarding the expression of *ZmTIP1;1*, the most remarkable results was that plants singly inoculated with the AM fungus reached the highest expression level under optimal conditions (Figure 7A). The application of the combined D + T stress significantly inhibited the expression of this gene in AM plants, while, on the contrary, the stress upregulated the expression of this gene in plants dually inoculated with AM + Bm.

The expression of the fungal aquaporin genes *GintAQP1* and *GintAQPF2* did not show significant differences among treatments. The expression of *GintAQPF1* was upregulated significantly (by 376%) in single AM plants upon exposure to the combined D + T stress, while, in plants dually inoculated with AM + Bm, the changes were not statistically significant (Figure 7B).

### 2.9. Accumulation of Aquaporins and Phosphorylation Status

The abundance of the PIP2 aquaporins analyzed (*ZmPIP2;1/2;2*, *ZmPIP2;4* and *ZmPIP2;5*) and the PIP2 phosphorylation status at Ser-280 (PIP2A), Ser-283 (PIP2B) and Ser-280 + Ser-283 (PIP2C), followed a similar trend in all the treatments. Only the inoculation with *B. megaterium* had a significant effect on these proteins, increasing their accumulation under optimal conditions and, in some cases, also when subjected to combined D + T stress, as compared to the rest of treatments (see Appendix A).

### 2.10. Hormone Accumulation in Sap

The sap ABA content increased significantly as a consequence of the combined D + T stress in uninoculated control plants and in plants dually inoculated with AM + Bm (Figure 8A). In plants singly inoculated with Bm or with the AM fungus, the increase due to the combined D + T stress was not significant. In the case of JA, its sap content was increased by the combined D + T stress only in uninoculated control plants, while, in plants singly inoculated with Bm, it decreased (Figure 8B). In the same way, the sap JA-Ile content was considerably reduced by the combined D + T stress in plants singly inoculated by Bm and also in uninoculated control plants (Figure 8C). The sap SA content was not affected by the microbial treatment when plants were cultivated under optimal conditions and only decreased significantly due to the combined D + T treatment in plants inoculated with Bm and in plants dually inoculated with AM + Bm (Figure 9A). Finally, the sap IAA content was only affected by the combined D + T stress treatment in uninoculated control plants that considerably enhanced its content (Figure 9B).

## 3. Discussion

In the last years, climate change has increased temperatures and altered precipitation regimes, which has led to a combined drought and high temperature stress in many areas and therefore a serious decline in crop production, especially for high-water-requiring crops such as maize [53].

Several studies have demonstrated that AM fungi alleviate stresses with an osmotic component such as drought, salinity or extreme temperatures by means of a combination of physical, physiological, nutritional and molecular effects [3,16,17,54]. Besides AM fungi, PGPR can also play important roles in host-plant health and alleviation of abiotic stresses. PGPRs help to mitigate stress effects at the whole-plant level [8]. They employ antioxidant mechanisms, improve root and shoot morphology, produce biofilms to improve water availability for the plant, increase water sustainability and produce secondary metabolites to benefit plant fitness [8,12,55,56].

Given the increased severity of climate change in recent years and the positive effects of AM fungi and PGPR on alleviation of abiotic stress in several plant species [44], this study aimed to examine the effects of the dual inoculation with the AM fungus *R. irregularis* and the PGPR *B. megaterium* on the tolerance of maize plants to combined drought and high temperature stress. The results from the present study show that dual inoculation with AM + Bm is more effective against the combined D + T stress than the single inoculation of these microorganisms. Thus, the highest shoot dry weight was achieved in AM + Bm plants, while the single Bm inoculation did not improve plant growth as compared to uninoculated controls plants. Moreover, the decrease in shoot dry weight due to combined D + T stress was lower in AM + Bm treatment (7.7%) than in single AM treatment (9.5%), uninoculated control plants or in single Bm-inoculated plants (decrease by 16.0% in both cases). This result was parallel to the result on relative electrolyte leakage, which was enhanced by the stress much less in dually inoculated AM + Bm plants than in uninoculated controls or in plants singly inoculated with each microorganism. The efficiency of phytosystem II and the photosynthetic activity also followed a similar trend and were less negatively affected by the combined D + T stress in dually inoculated AM + Bm plants than in the rest of treatments. It is noteworthy that the high photosynthetic activity, efficiency of photosystem II and stomatal conductance of these plants were positively correlated with the high levels of Ja-Ile in sap of this treatment (see Appendix A).

Drought and heat stress are simultaneously prevalent in semi-arid areas. Their combined effects on plant physiology are still little known, but it is thought that are more adverse for plants than each stress separately [4,57]. For instance, when tobacco plants were subjected to drought and heat stress, they exhibited a higher leaf temperature as compared to plants subjected only to heat stress, due to the precedence and prevalence of stomatal closure to reduce water loss over the need to cool the leaves by keeping them open [58]. In the same way, accumulation of proline is one of the major osmoprotectants in plants subjected to drought stress, whereas under combined stress conditions, proline accumulation is negative for the plant and sucrose is the main osmoprotectant that accumulates [59]. It is also known that the concurrence of both stresses can lead to increased leaf temperature, decreased stomatal conductance, diminished concentrations of photosynthetic pigments, impairment of photosystem II and reduced RuBisCO activity and net photosynthesis [8]. All of these processes will negatively affect plant productivity and are responsible for the reduced plant growth. In this study, the dual inoculation with AM + Bm improved most of these parameters, which are surely involved in the better performance and growth parameters of these dually inoculated plants under the combined D + T stress. For instance, the higher efficiency of photosystem II in dually inoculated plants indicates that the photochemical apparatus of these plants did not lose functionality in light conversion [60,61]. The dual AM + Bm inoculation also significantly reduced the membrane electrolyte leakage, which also contributed to the better performance of these plants. Indeed, the membrane electrolyte leakage is an estimation of cell membrane stability and is an index of the tolerance to the stress imposed [31,62] since a higher membrane stability often correlates with a lower lipid peroxidation [63].

Regarding root hydraulic properties, both Lo and Lpr reached the maximum values in dually inoculated AM + Bm plants, both under well-watered and under combined D + T stress, and the stress applied did not negatively affect Lo in AM + Bm plants, while it decreased Lo in the rest of treatments, reaching its lowest value in uninoculated control plants subjected to D + T stress. Lo measures water flow through the cell-to-cell pathway in which aquaporins are involved. The maintenance of Lo in dually inoculated AM + Bm plants could be related to an increased expression of plant or fungal aquaporins [30,34]. Indeed, plant exposure to both drought and high temperatures triggers physiological and molecular changes that affect the aquaporin expression patterns and root hydraulics, which is directly related to aquaporins [14,64,65]. In the case of drought stress, both the down- and upregulation of aquaporins have been described [14]. In the case of heat stress, generally the long-term stress response is a decrease in aquaporins expression. However, different plants can respond differently to different intensities and times of stress exposure. For instance, in soybeans, several PIPs and TIPs were upregulated in roots and leaves during the first six hours after heat treatment. In contrast, after 12 h of heat stress, all were downregulated in leaves [66]. Some plants, such as *Rhazya stricta* L., have shown an adaptation to the hottest daylight hours based on increased PIP aquaporins expression in leaves [67], similar to the strawberry (*Fragraria vesca* L.), which increases PIP expression after one-hour of heat stress [68]. On the contrary, in tobacco plants, there is a decrease in the PIP2s levels in roots after 50 °C treatment [69]. Moreover, the pattern of aquaporin gene expression can be different in response to heat stress among cultivars of the same plant species, as described in *Setaria italic* L. [65].

Curiously, in this study, some of the aquaporins analyzed (*ZmPIP1;3*, *ZmTIP1.1* and *GintAQPF1*) exhibited a higher gene expression in dually inoculated AM + Bm plants than in the other treatments when subjected to combined D + T stress, while *ZmPIP2;2* was downregulated in this treatment. The aquaporin *ZmPIP1;3*, like other PIP1s, can interact with PIP2s in order to regulate their water transport capacity [70]. *ZmTIP1;1* is the most expressed TIP in maize [18] and can transport water and other compounds, such as, urea, ammonia, boron or H_2_O_2_ [29,31]. The fungal aquaporin *GintAQPF1* was shown to be able to transport water and to be expressed both in extraradical mycelium and in maize cortical cells containing arbuscules. Its expression was upregulated by drought stress [71], as in our study. Finally, *ZmPIP2;2* has a high capacity for water transport in *Xenopus laevis* oocytes [29]. Thus, such tight regulation of aquaporin expression makes sense in the context of a fine control of water balance in maize roots dually inoculated with AM + Bm.

On the other hand, aquaporin abundance in root cortex cells may alter Lo, especially during water shortage, where aquaporins are thought to be regulated for the maintenance of the adequate water balance [19] Thus, the maintenance of Lo in dually inoculated AM + Bm plants may be due to additional mechanisms, such as increased abundance and/or activity of the aquaporins due to post-translational modifications [34,72]. PIPs were proved to contribute to the adaptation of plants to drought episodes, also contributing to rehydration of the whole plant after water shortage [73] and, in this study, we examined the abundance of several PIPs and the phosphorylation status of PIP2s. However, their abundances and phosphorylation status were only significantly affected by inoculation with Bm, but not in the dual inoculation, and it is likely that his is not the reason for the maintenance of Lo in dually inoculated AM + Bm plants. Alternatively, it may be also due to changes in the plant hormonal content [74]. Indeed, plant hormones such as ABA; JA and its derivative, JA-Ile; SA; or IAA may well be involved in the regulation of the own plant–microbe interaction [43,48], in the activity of the plant aquaporins or in posttranscriptional changes in these aquaporins, and, in turn, this alters their water channel activity [30,45,75,76]. In this study, the plants dually inoculated with AM + Bm and subjected to combined D + T stress also contained the highest ABA and JA-Ile contents in sap, while hormones such as SA and IAA were low in comparison to uninoculated control plants.

Experimental evidence suggests that, in AM plants, the modulation of ABA, auxins and/or SA levels may contribute to switching between apoplastic and cell-to-cell water pathways [30,32,77]. Indeed, ABA has been identified as a possible aquaporin regulator [75,78] and increases Lo in maize, lettuce or wheat plants [79,80,81]. MeJA was also shown to increase Lo in tomato, bean or Arabidopsis in a calcium- and ABA-dependent way [30]. IAA inhibited the expression of most PIPs in Arabidopsis [82], and, in maize, IAA reduced Lo in an aquaporin-mediated mode [35]. In a similar way, SA downregulates PIP aquaporins and root hydraulic conductivity by an ROS-mediated mechanism which provoked membrane internalization of PIP [78], and this was also found in maize [32]. Thus, enhanced ABA and JA-Ile contents in AM + Bm plants and reduced SA and IAA contents may have contributed to maintain high levels of Lo in these plants. Indeed, ABA and JA-Ile contents had a positive correlation with Lo and Lpr in this study, while SA and IAA contents had a negative correlation with Lo and Lpr (see Appendix A). Moreover, ABA had a positive correlation with the expression of *ZmPIP1;3* aquaporin.

In conclusion, the data obtained in this study show that that the single inoculation with the AM fungus *R. irregularis* was more effective than the single inoculation with the PGPR *B. megaterium* in promoting maize growth and performance both under optimal conditions and when subjected to combined D + T stress. However, the dual inoculation with AM + Bm was more effective against a combined D + T stress than the single inoculation of these microorganisms. The positive effect of the dual inoculation was related to a synergistic enhancement of the efficiency of phytosystem II, stomatal conductance and photosynthetic activity, as well as a reduction of membrane electrolyte leakage. At the same time, the dual inoculation of AM + Bm allowed plants to maintain higher values of Lo and Lpr under stress conditions, and this was related to the upregulation of the aquaporins *ZmPIP1;3*, *ZmTIP1.1* and *GintAQPF1* and downregulation of *ZmPIP2;2*, as well as to enhanced sap ABA and JA-Ile contents and reduced sap SA and IAA contents. The present results demonstrate that, at least under controlled conditions, combining beneficial soil microorganisms such as the AM fungus *R. irregularis* and the PGPR *B. megaterium* is a useful and sustainable tool to improve plant productivity under combined drought and high temperature stress. Future studies are needed to check this conclusion under field conditions and to validate the use of these microorganisms to improve crop productivity under the current climate-change scenario that causes the combination of drought and enhanced temperatures.

## 4. Materials and Methods

### 4.1. Design of the Experiment and Statistical Analysis

The experiment consisted of a factorial design with two factors: (1) Microbial inoculation treatment, with non-inoculated control plants (C), plants inoculated with the AM fungus *Rhizophagus irregularis*, strain EEZ 58 (AM); plants inoculated with the plant growth promoting rhizobacteria *Bacillus megaterium* (Bm); and plants dually inoculated with *R. irregularis* and *B. megaterium* (AM + B). (2) Stress treatment, so that one half of the plants were cultivated under well-watered (WW) conditions and standard temperature throughout the entire experiment (these are considered optimal conditions in this study) and the other half of the plants were subjected to combined drought and high temperature (5 °C above standard temperature) stresses (D + T) for 15 days before harvest. The different combinations of these factors gave a total of 8 treatments. Fifteen replicates were used for each treatment, giving a total of 120 plants.

The data were tested for a normal distribution and variance homogeneity (*p* < 0.05), and, when needed, variables were log transformed before further analyses. The data were subjected to a two-way analysis of variance (two-way ANOVA), with inoculation treatment, water regime and inoculation-treatment–water-regime interaction as sources of variation. Post hoc comparisons with the Duncan’s test were used to find out differences between groups. The SPSS Statistics (Version 27, IBM Analytics, Armonk, NY, USA) was used to perform parametric data analysis. When the data did not fit the normality distribution, Kruskal–Wallis as a non-parametric test of variance was applied. In these variables, Tukey–Kramer was used as a post hoc test. Correlations between the different parameters were performed by calculating the Pearson correlation coefficients (Appendix A). The analyses of non-parametric variables and Pearson correlation were performed using the JMP^®^, Version 10 (SAS Institute Inc., Cary, NC, USA, 1989–2007).

### 4.2. Soil and Biological Materials

The soil used was a loam and was collected on 1 September 2021 at the grounds of IFAPA (Granada, Spain), sieved (2 mm), diluted with quartz sand (<1 mm) (1:1, soil:sand, *v*/*v*) and sterilized by steaming (100 °C for 1 h on 3 consecutive days). The soil pH was 8.1 (water) and contained 0.85% organic matter with the following nutrient concentrations (mg kg^−1^): N, 1; P, 10 (NaHCO_3_-extractable P); and K, 110. The soil texture was made of 38.3% sand, 47.1% silt and 14.6% clay.

Seeds of maize (*Zea mays* L.), cultivar PR34B39, were provided by Pioneer Hi-Bred, Spain (DuPont Pioneer Corporation, Johnston, IA, USA). Seeds were pre-germinated on moist sand for 5 days (from 7 to 12 September 2021) and then transferred to pots of 1.5 L capacity containing 1250 g of the soil/sand mixture described above on 12 September 2021. A total of 120 pots were prepared, each one containing one maize seedling.

Mycorrhizal inoculum was bulked in an open-pot culture of *Z. mays* L. and consisted of soil, spores, mycelia and infected root fragments. The AM fungus was *Rhizophagus irregularis* (Schenck and Smith), belonging to the Zaidin Experimental Station (EEZ) Collection, strain EEZ 58. Ten grams of inoculum with about 65 infective propagules per gram (according to the most probable number test) was added to the appropriate pots at the time of seedlings transplantation to pots (12 September 2021). Non-inoculated control plants received the same amount of autoclaved mycorrhizal inoculum together with a 3 mL aliquot of a filtrate (<20 µm) of the AM inoculum in order to provide a general microbial population free of AM propagules.

Appropriate pots were inoculated with a *Bacillus megaterium* strain isolated and tested in previous studies [36,38,83,84]. For that, two days before starting this experiment, *B. megaterium* strain was grown in nutrient broth medium for 48 h at 28 °C and then centrifuged at 4500× *g* for 5 min. The pellet was suspended in sterilized water. One milliliter of the suspension containing 10^8^ cfu mL^–1^ was added to each pot at the time of the seedlings’ transplantation to pots (12 September 2021) and repeated seven days later (19 September 2021).

### 4.3. Growth Conditions

The experiment was conducted in a greenhouse with the following conditions: 16/8 h light/dark period, a relative humidity of 50–60% and standard temperatures of 19/24 °C (night/day) or high temperatures of 24/29 °C (night/day) in the case of treatments subjected to stress. The average photosynthetic photon flux density was 800 µmol m^−2^ s^−1^, as measured with a light meter (LICOR, Lincoln, NE, USA, model LI-188B). Plants were cultivated for a total of 8 weeks (from 12 September 2021 to 14 November 2021), and 4 weeks after sowing, all plants started receiving 10 mL per pot and per week of Hoagland nutrient solution [85] containing only 25% of P in order to provide basic nutrients but avoiding inhibition of AM symbiosis due to a high P application.

Soil moisture was controlled with a ML2 ThetaProbe (AT Delta-T Devices Ltd., Cambridge, UK). Thus, during the first 6 weeks after sowing (from 12 September 2021 to 29 October 2021), water was daily supplied to maintain soil at 100% of field capacity in all treatments. A previous experiment using a pressure plate apparatus showed that the 100% soil water-holding capacity corresponds to 22% volumetric soil moisture measured with the ThetaProbe. Then half of the plants (unstressed plants) were maintained in the same greenhouse under the abovementioned conditions during the entire experiment (from 12 September 2021 to 14 November 2021). These were considered optimal conditions in this study. The other half of the plants (stressed plants) were moved to a parallel greenhouse, where the temperature was set up 5 °C above standard (to reach 24/29 °C, night/day temperatures). The plants subjected to high temperature were also allowed to dry until the soil water content reached 60% of field capacity (one day needed). The 60% of soil water-holding capacity corresponds to 7% volumetric soil moisture measured with the ThetaProbe (also determined previously with a pressure plate apparatus). The soil water content was measured daily with the ThetaProbe ML2 before rewatering (at the end of the afternoon), reaching a minimum soil water content around 55% of field capacity in the drought-stressed and high-temperature-stressed treatments. The amount of water lost was daily replaced to each pot in order to keep the soil water content at the desired levels of either 7% (stressed plants) or 22% (non-stressed plants) of volumetric soil moisture [86]. Plants were maintained under such conditions for 15 additional days before harvesting (from 30 October to 14 November 2021).

### 4.4. Measurements

#### 4.4.1. Biomass Production, Shoot Water Content and Symbiotic Development

At harvest (8 weeks after sowing), the shoot and root systems of fifteen replicates per treatment were separated and weighed to determine fresh weights (FWs). Subsequently, the dry weight (DW) was also measured after drying in a forced hot-air oven at 70 °C for 2 days. The shoot water contents (WCs) were determined using the following equation: WC (%) = [(FW − DW)/FW] × 100.

The percentage of mycorrhizal root colonization was estimated by visual observation according to Phillips and Hayman [87], and the extent of mycorrhizal colonization was quantified according to the gridline intersect method [88] in five replicates per treatment.

#### 4.4.2. Membrane Electrolyte Leakage

Six plants per treatment were used to determine leaf electrolyte leakage. Samples were first washed with deionized water to remove surface-adhered electrolytes. Then the samples were placed in closed vials containing 10 mL of deionized water and incubated at 25 °C on a rotary shaker (at 100 rpm) for 3 h. The electrical conductivity of the solution (L0) was determined using a conductivity meter (Metler Toledo AG 8603, Greifensee, Switzerland). After that, samples were placed at −80 °C for 2 h. Subsequently, tubes were incubated again at room temperature under smoothly agitation for 3 h, and the final electrical conductivity (Lf) was measured. The electrolyte leakage was quantified as follows: [(L0 − Lwater)/(Lf − Lwater)] × 100, with Lwater being the conductivity of the deionized water used to incubate the samples.

#### 4.4.3. Efficiency of Photosystem II

The efficiency of photosystem II was measured with a FluorPen FP100 (Photon Systems Instruments, Brno, Czech Republic), which allows a non-invasive assessment of plant photosynthetic performance by measuring chlorophyll a fluorescence. FluorPen quantifies the quantum yield of photosystem II as the ratio between the actual fluorescence yield in the light-adapted state (FV′) and the maximum fluorescence yield in the light-adapted state (FM′), according to Oxborough and Baker [89]. Measurements were taken in the second youngest leaf of ten different plants of each treatment.

#### 4.4.4. Gas Exchange Measurements

After 8 weeks of plant cultivation, the net photosynthesis (An), stomatal conductance (gs), and instantaneous water-use efficiency (iWUE = An/gs) of fully expanded young leaves in eight different plants per treatment were measured using a portable photosystem system LI-6400 (LICOR Biosciences, Lincoln, NE, USA) two hours after sunrise. Measurements were performed at an ambient CO_2_ concentration of 390 μmol m^−2^, temperature of 25/30 °C, 50 ± 5% relative humidity and a PPFD of 1000 μmol m^−2^s^−1^.

#### 4.4.5. Osmotic Root Hydraulic Conductivity (Lo)

Before harvest, the osmotic root hydraulic conductivity (Lo) was measured on detached roots exuding under atmospheric pressure for two hours [27]. Under these conditions, water is only moving following an osmotic gradient. Therefore, the water would be moving through the cell-to-cell path [90]. Eight plants per treatment were used for this determination. Lo was calculated as Lo = Jv/∆Ψ, where Jv is the exuded sap flow rate, and ∆Ψ is the osmotic potential difference between the exuded sap and the nutrient solution where the pots were immersed. These measurements were carried out 3 h after the onset of light in eight different plants per treatment.

#### 4.4.6. Hydrostatic Root Hydraulic Conductivity (Lpr)

Lpr was determined at noon in seven plants per treatment with a Scholander pressure chamber, as described by Bárzana et al. [28]. A gradual increase of pressure (0.3, 0.4 and 0.5 MPa) was applied at 2-minute intervals to the detached roots. Sap was collected at the three pressure points in seven different plants per treatment. Sap flow was plotted against pressure, with the slope being the root hydraulic conductance (L) value. Lpr was determined by dividing L by root dry weight (RDW) and expressed as mg H_2_O g RDW^−1^ MPa^−1^ h^−1^.

#### 4.4.7. Quantitative Real-Time RT-PCR

Three biological replicates of maize roots were used to extract total RNA, as described in Quiroga et al. [31]. First-strand cDNA was synthesized using 1 µg of purified RNA with the Maxima H Minus first strand cDNA synthesis kit (Thermo Scientific™, Waltham, MA, USA), following the manufacturers’ instructions.

The expression of eight previously selected maize aquaporins (*ZmPIP1;1*, *ZmPIP1;3*, *ZmPIP2;2*, *ZmPIP2;4*, *ZmTIP1;1*, *ZmTIP2;3*, *ZmTIP4;1* and *ZmNIP2;1*) [31] was measured by qRT-PCR using 1 µL of diluted cDNA (1:9) with PowerUpTM SYBRTM Green Master Mix in a QuantStudioTM 3 system (Thermo Fisher Scientific, Waltham, MA, USA). The reaction was repeated for 40 cycles, at an annealing temperature of 58 °C, for all primers. Four reference genes were measured in all the treatments for the normalization of gene-expression values. These genes were poliubiquitin (gi:248338), tubulin (gi:450292), GAPDH (gi:22237) and elongation factor 1 (gi:2282583) [29]. Standardization was carried out based on the expression of the two best-performing reference genes under our specific conditions, which were chosen by using the “NormFinder” algorithm [91] (https://moma.dk/normfinder-software (accessed on 2 February 2023)). Thus, expression levels were normalized according to Zmtubulin and ZmGAPDH genes. Fungal aquaporins (*GintAQP1*, *GintAQPF1* and *GintAQPF2*) were analyzed as previously described [71,92], using fungal elongation factor 1α (Accession No. DQ282611) as reference gene for standardization. The relative abundance of transcripts was calculated using the 2^−ΔΔct^ method [93]. The threshold cycle (Ct) of each biological sample was determined in duplicate. Negative controls without cDNA were used in all PCR reactions.

#### 4.4.8. Aquaporins Abundance and PIP2s Phosphorylation Status

Sub-cellular fractionation was performed according to Hachez et al. [94], with slight modifications. Pieces of intact roots were grinded with 6 mL of a protein extraction buffer containing 250 mM Sorbitol, 50 mM Tris-HCl (pH 8), 2 mM EDTA and protease inhibitors. All steps were performed at 4 °C. The homogenate was centrifuged for 10 min at 770× *g*, and the supernatant obtained was centrifuged 10 min at 10,000× *g*. The resulted supernatant was finally centrifuged for 30 min at 100,000× *g*, and the final pellet (corresponding to the microsomal fraction) was resuspended in 20 µL of suspension buffer (5 mM KH_2_PO_4_, 330 mM sucrose, 3 mM KCl, pH 7.8) and sonicated twice for 5 s. Total protein amounts were quantified by Bradford analysis, and the abundance of specific proteins was measured by ELISA. A 2 µg aliquot of microsomal fraction was incubated at 4 °C overnight in carbonate/bicarbonate coating buffer at pH 9.6. The next day, proteins were cleaned by 3 × 10 min washes with Tween Tris-buffered saline solution (TTBS) and blocked with 1% bovine serum albumin (BSA) on TTBS 1 h at room temperature. After three more washes with TTBS, proteins were incubated with 100 µL of the primary antibody (1:1000 in TTBS *v*/*v*) for 1 h at room temperature.

We used primary antibodies recognizing several PIPs’ aquaporins, namely *ZmPIP2;1/2;2*, *ZmPIP2;4* and *ZmPIP2;5* [94], as well as three antibodies that recognize the phosphorylation of PIP2 proteins in the C-terminal region: PIP2A (Ser-280), PIP2B (Ser-283) and PIP2C (Ser-280/Ser-283) [77]. A goat anti-rabbit IgG coupled to horseradish peroxidase (Sigma-Aldrich Co., St. Louis, MO, USA) was used as secondary antibody at 1:10,000.

#### 4.4.9. Sap Hormonal Content

In sap, IAA, ABA, SA, JA and JA-Ile were analyzed according to Albacete et al. [95], with some modifications. Briefly, xylem sap samples from eight different plants per treatment were filtered through 13 mm diameter Millex filters with a 0.22 µm pore size nylon membrane (Millipore, Bedford, MA, USA). Then, 10 µL of filtrated extract was injected in a U-HPLC-MS system consisting of an Accela Series U-HPLC (ThermoFisher Scientific, Waltham, MA, USA) coupled to an Exactive mass spectrometer (ThermoFisher Scientific), using a heated electrospray ionization (HESI) interface. Mass spectra were obtained using Xcalibur software version 2.2 (ThermoFisher Scientific). For quantification of the plant hormones, calibration curves were constructed for each analyzed component (1, 10, 50 and 100 µg L^−1^).

## Figures and Tables

**Figure 1 ijms-24-05193-f001:**
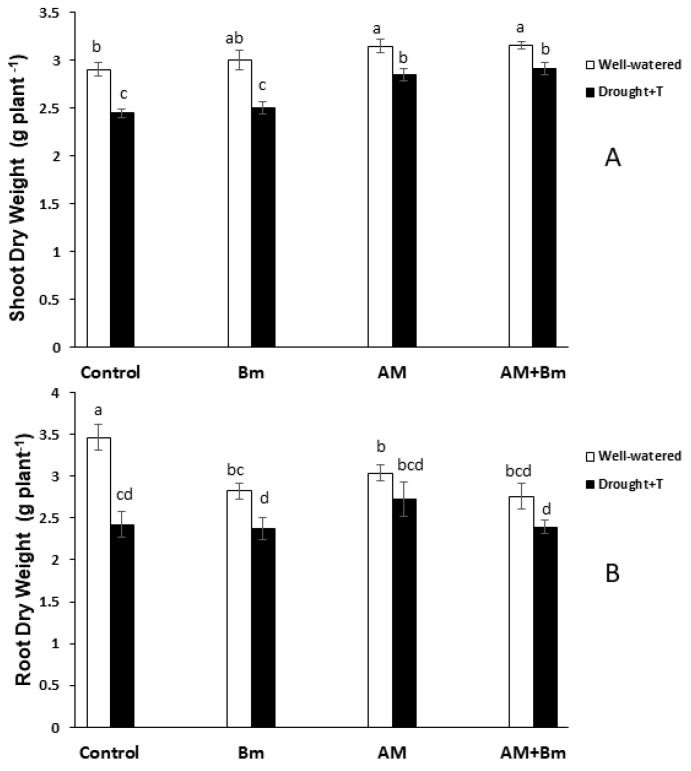
(**A**) Shoot dry weight and (**B**) root dry weight in maize plants inoculated or not (control) with a PGPR strain of *Bacillus megaterium* (Bm), with the arbuscular mycorrhizal fungus *Rhizophagus irregularis* (AM) or with both microorganisms (AM + Bm). Plants were cultivated under standard ambient temperature and well-watering conditions (well-watered) or subjected to a combined drought and high temperature stress (drought + T) for 15 days before harvest. Data represent the means (n = 15) ± S.E. Different letters indicate significant differences between treatments (*p* < 0.05) based on Duncan’s test.

**Figure 2 ijms-24-05193-f002:**
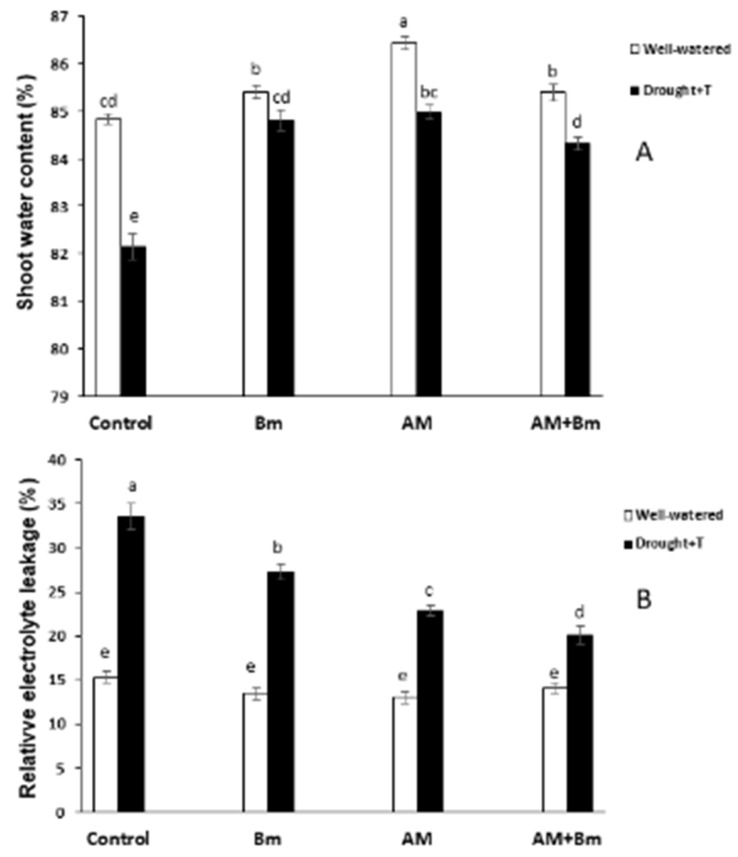
(**A**) Shoot water content and (**B**) relative electrolyte leakage in maize plants inoculated or not (control) with a PGPR strain of *Bacillus megaterium* (Bm), with the arbuscular mycorrhizal fungus *Rhizophagus irregularis* (AM) or with both microorganisms (AM + Bm). Plants were cultivated under standard ambient temperature and well-watering conditions (well-watered) or subjected to a combined drought and high temperature stress (drought + T) for 15 days before harvest. Data represent the means (n = 15) (shoot water content) or (n = 6) (relative electrolyte leakage) ± S.E. Different letters indicate significant differences between treatments (*p* < 0.05) based on Duncan’s test.

**Figure 3 ijms-24-05193-f003:**
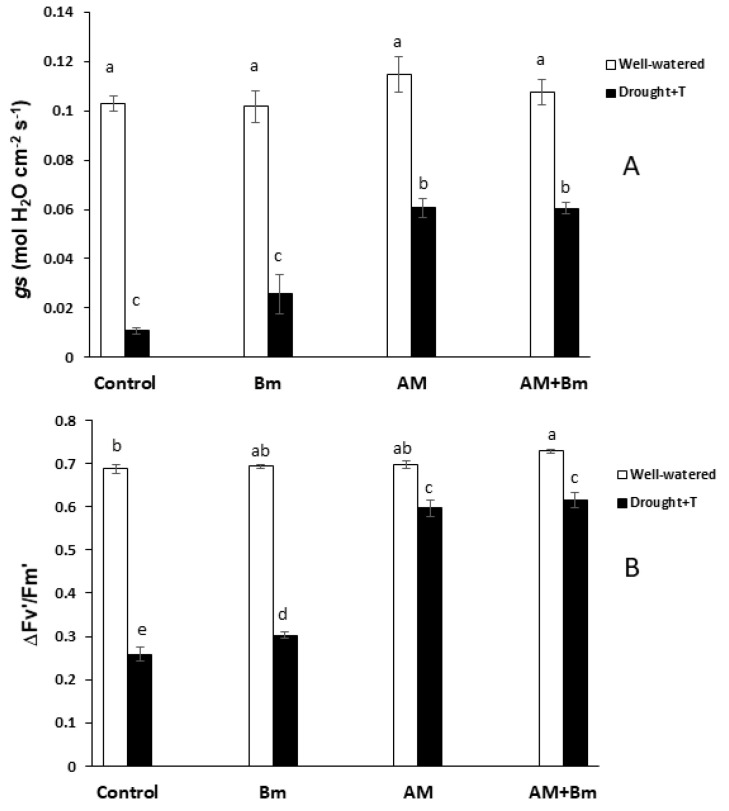
(**A**) Stomatal conductance (gs) and (**B**) photosystem II efficiency in the light-adapted state (ΔFv/Fm′) in maize plants inoculated or not (control) with a PGPR strain of *Bacillus megaterium* (Bm), with the arbuscular mycorrhizal fungus *Rhizophagus irregularis* (AM) or with both microorganisms (AM + Bm). Plants were cultivated under standard ambient temperature and well-watering conditions (well-watered) or subjected to a combined drought and high temperature stress (drought + T) for 15 days before harvest. Data represent the means (n = 8) (gs) or (n = 10) (ΔFv/Fm’) ± S.E. Different letters indicate significant differences between treatments (*p* < 0.05) based on Tukey–Kramer test.

**Figure 4 ijms-24-05193-f004:**
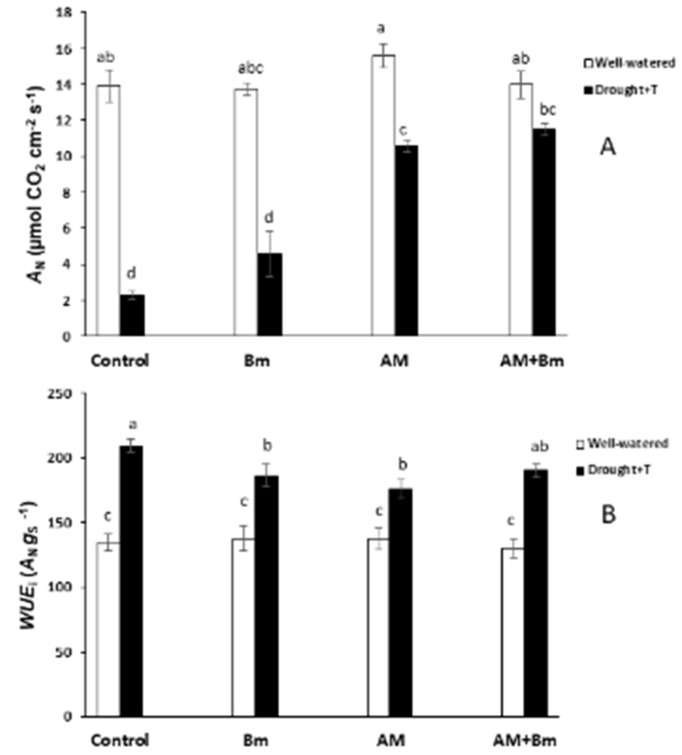
(**A**) Net photosynthetic activity (An) and (**B**) instantaneous water-use efficiency (WUEi) in maize plants inoculated or not (control) with a PGPR strain of *Bacillus megaterium* (Bm), with the arbuscular mycorrhizal fungus *Rhizophagus irregularis* (AM) or with both microorganisms (AM + Bm). Plants were cultivated under standard ambient temperature and well-watering conditions (well-watered) or subjected to a combined drought and high temperature stress (drought + T) for 15 days before harvest. Data represent the means (n = 8) ± S.E. Different letters indicate significant differences between treatments (*p* < 0.05) based on the Tukey–Kramer test.

**Figure 5 ijms-24-05193-f005:**
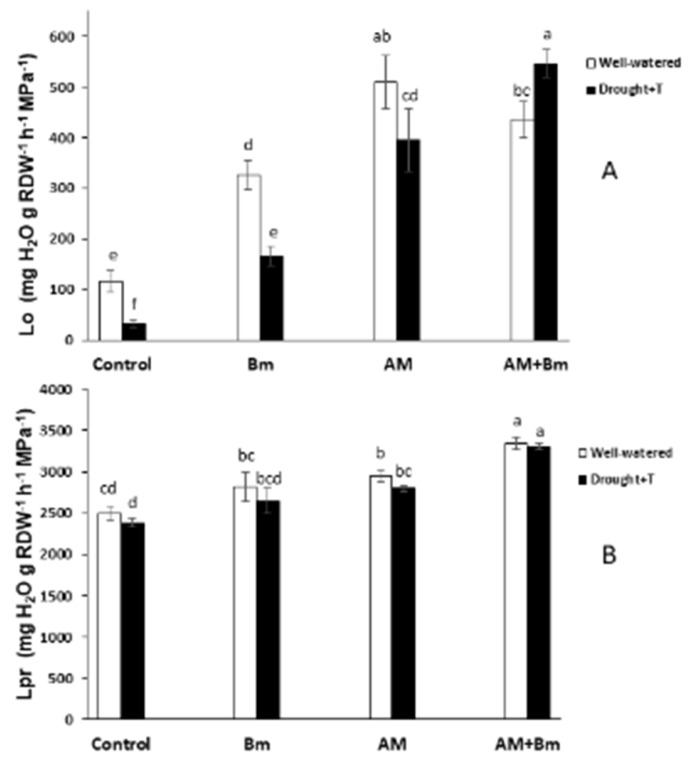
(**A**) Osmotic root hydraulic conductivity (*Lo*) and (**B**) hydrostatic root hydraulic conductivity (*Lpr*) in maize plants inoculated or not (control) with a PGPR strain of *Bacillus megaterium* (Bm), with the arbuscular mycorrhizal fungus *Rhizophagus irregularis* (AM) or with both microorganisms (AM + Bm). Plants were cultivated under standard ambient temperature and well-watering conditions (well-watered) or subjected to a combined drought and high temperature stress (drought + T) for 15 days before harvest. Data represent the means (n = 8) (*Lo*) or (n = 7) (*Lpr*) ± S.E. Different letters indicate significant differences between treatments (*p* < 0.05) based on Duncan’s test.

**Figure 6 ijms-24-05193-f006:**
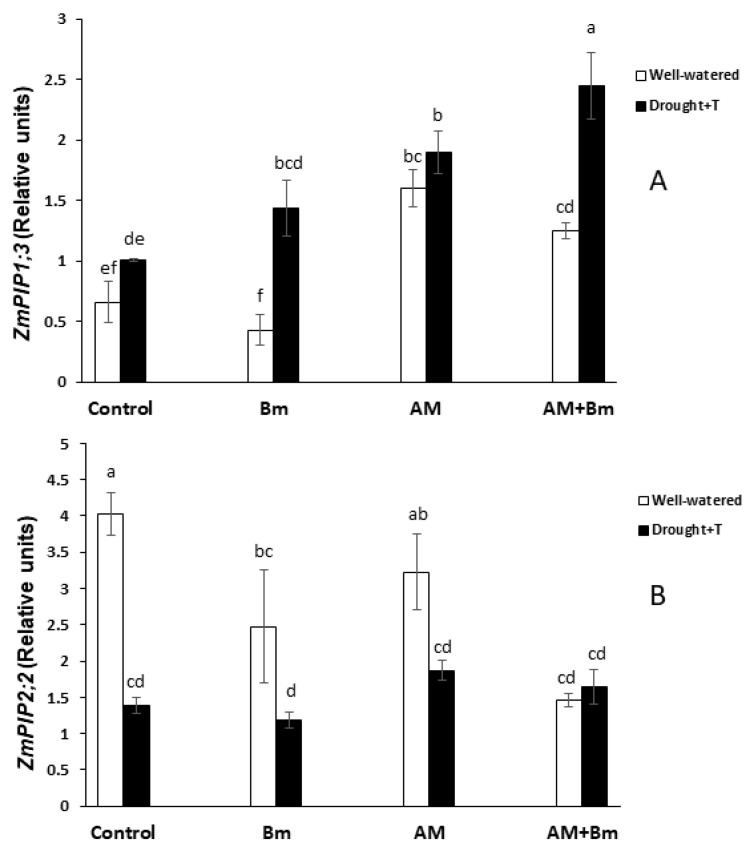
Expression of *ZmPIP1;3* (**A**) and *ZmPIP2;2* (**B**) in roots of maize plants inoculated or not (control) with a PGPR strain of *Bacillus megaterium* (Bm), with the arbuscular mycorrhizal fungus *Rhizophagus irregularis* (AM) or with both microorganisms (AM + Bm). Plants were cultivated under standard ambient temperature and well-watering conditions (well-watered) or subjected to a combined drought and high temperature stress (drought + T) for 15 days before harvest. Data represent the means (n = 3) ± S.E. Different letters indicate significant differences between treatments (*p* < 0.05) based on Duncan’s test.

**Figure 7 ijms-24-05193-f007:**
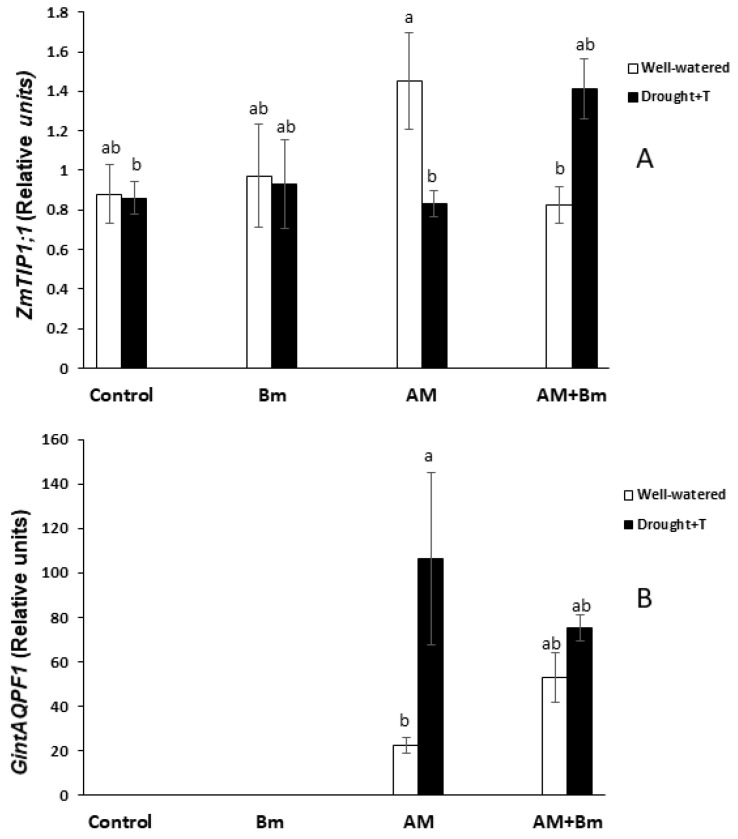
Expression of *ZmTIP1;1* (**A**) and *GintAQPF1* (**B**) in roots of maize plants inoculated or not (control) with a PGPR strain of *Bacillus megaterium* (Bm), with the arbuscular mycorrhizal fungus *Rhizophagus irregularis* (AM) or with both microorganisms (AM + Bm). Plants were cultivated under standard ambient temperature and well-watering conditions (well-watered) or subjected to a combined drought and high temperature stress (drought + T) for 15 days before harvest. Data represent the means (n = 3) ± S.E. Different letters indicate significant differences between treatments (*p* < 0.05) based on Duncan’s test.

**Figure 8 ijms-24-05193-f008:**
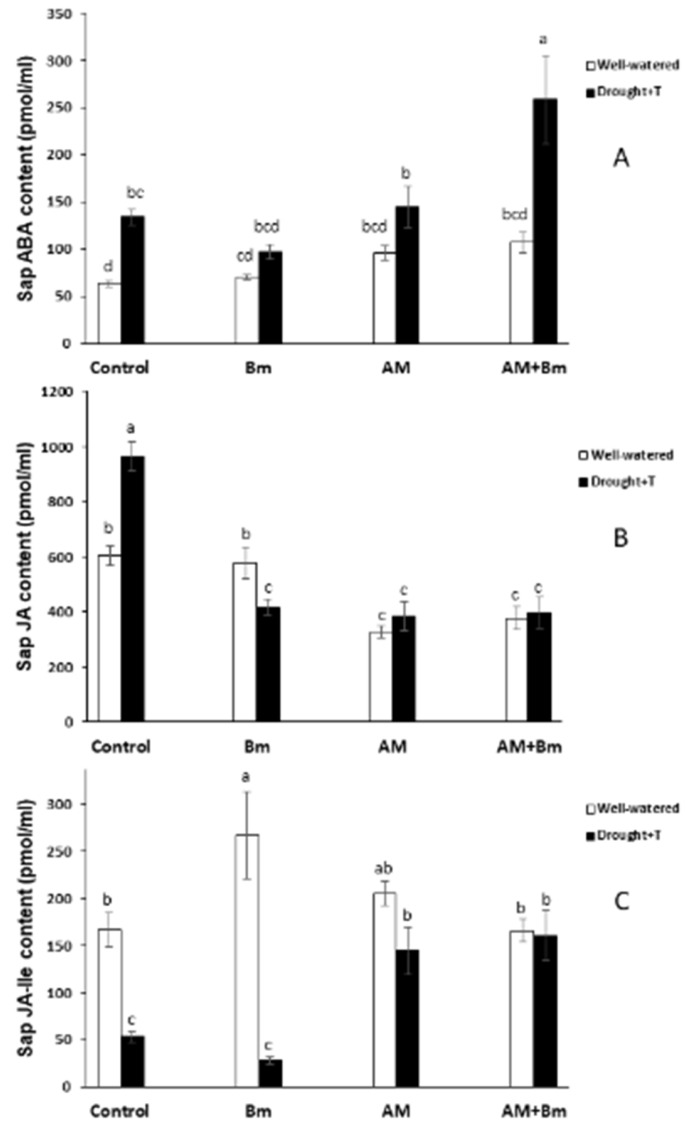
(**A**) ABA, (**B**) JA and (**C**) JA-Ile contents in sap of maize plants inoculated or not (control) with a PGPR strain of *Bacillus megaterium* (Bm), with the arbuscular mycorrhizal fungus *Rhizophagus irregularis* (AM) or with both microorganisms (AM + Bm). Plants were cultivated under standard ambient temperature and well-watering conditions (well-watered) or subjected to a combined drought and high temperature stress (drought + T) for 15 days before harvest. Data represent the means (n = 8) ± S.E. Different letters indicate significant differences between treatments (*p* < 0.05) based on Duncan’s test or Tukey–Kramer test in the case of JA-Ile.

**Figure 9 ijms-24-05193-f009:**
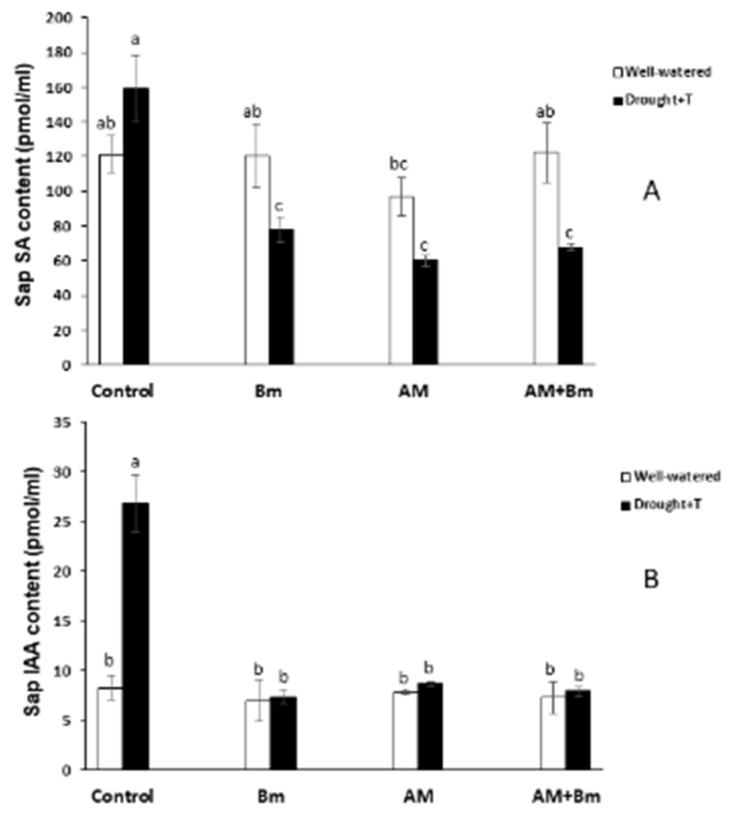
(**A**) SA and (**B**) IAA contents in sap of maize plants inoculated or not (control) with a PGPR strain of *Bacillus megaterium* (Bm), with the arbuscular mycorrhizal fungus *Rhizophagus irregularis* (AM) or with both microorganisms (AM + Bm). Plants were cultivated under standard ambient temperature and well-watering conditions (well-watered) or subjected to a combined drought and high temperature stress (drought + T) for 15 days before harvest. Data represent the means (n = 8) ± S.E. Different letters indicate significant differences between treatments (*p* < 0.05) based on Duncan’s test.

## Data Availability

Not applicable.

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
