# Peer review of "Dual Inoculation with Rhizophagus irregularis and Bacillus megaterium Improves Maize Tolerance to Combined Drought and High Temperature Stress by Enhancing Root Hydraulics, Photosynthesis and Hormonal Responses"

_ijms, 2023, doi:10.3390/ijms24065193_

Round 1

Reviewer 1 Report

The authors report the mitigation of heat-/drought-stress in maize with adding microorganisms in the soil. The manuscript may be published in the journal after revision. The points that should be taken into account for the revision are mentioned below, first for majors and then for minors.

Major issues

Date

The authors do not provide any date of the experiment. They should mention the date on which they, for example, collected the soil (L463), sowed seeds on pots (L469), prepared the pots (L470) and so on.

Statistical analyses: ANOVA

The authors applied ANOVAs of factorial design to this study (L445). This is misused. They hypothesized that two microorganisms would have been under the influence of the physical conditions (L17-20). This means that the hypotheses cannot be tested by an ANOVA for a factorial design which is used for two or more independent factors. The experimental design in their study should be examined by Nested ANOVAs. Also, as they mention at L22-23, the two bacteria could have been interactive, models to test the results should include the effect of each bacterium as well as the interaction term of these bacteria.

Since the authors tested their results with ANOVA, they should provide the results of the tests with statistics (F-values, dfs and probabilities).

The authors should conclude the effects of each bacterium and two under the influence of the ambient conditions (humidity and temperature).

Variance

Beside the issues on the experimental design, another serious problem seems to be involved in the results. As long as seen in all figures, the results do not follow “homogeneity of variances”. This is the most serious problem in the use of parametric analyses such as ANOVAs. The authors “must” check the conditions for the use of appropriate statistical tests that they applied to the data.

Duncan’s test

This should be avoided, as it is very likely to provide Type I error. Instead, other tests should be considered.

Unit for temperature

Temperatures should be indicated as “ºC”, not “º” underlined.

English

English should be checked by a native English speaker before resubmission.

Minor points

L15-16

The acronyms of two bacteria are shown in different manner. If one should be indicated for its function, like “AM”, the other should follow it, like PG. If one should be on its scientific name, the other should be too, like Ri.

Also, when any acronyms or abbreviations are referred to first, it should be spelled out.

Introduction

It is too long and must be shortened at least by two-thirds, or hopefully by half. Most or all sentences on L32-52 and L106-133 may be deleted without losing the importance of this study.

L177

The definition for “optimal” must be given in Materials and Methods.

L438-442

The authors conclude that the combination of the two bacteria can be used as an efficient measure in agriculture. This is too early. Since the study was done in a greenhouse where the plants were grown on pot, it still remains that these species can work in field. Their conclusion needs to wait until evidence from field experiments is obtained.

L448

Probably “(B)” is wrongly spelled.

L470

The authors should mention 1) the size of the pot, 2) the number of pots prepared and 3) the number of germinates planted on each pot. As mentioned in Major points, the date on which seed were sown and germinated should be provided.

L471-478

The authors should explain briefly when the two bacteria were collected originally and how the strains were established for the experiment.

L517

Explain replicate. In particular, how many replicates were set in each experiment should be described clearly.

L548

What does mean “after 8 weeks of growth”. Explain and give the definition for “growth”.

L580

Refer to the reference number, instead of the authors with the publication year.

L643

The authors use “value” for the means with se throughout this manuscript. It is not clear what the “value” indicates. Probably they refer to the number of replicates in the experiment. If so, value is not needed to write. It should be written just like, “data are shown in the means ± se.

Figures

Letters are too small. The letter should be larger.

Author Response

Please, see attached doccument

Reviewer 2 Report

This Ms presents interesting and useful findings. A few suggestions::

First, italicize all genus and species names. This is not done in some cases in both the body and the literature sections, including on line 141 where Abaridopsis needs to have the first letter capitalized.

Line 68: Reword to say "Among these beneficial are the arbuscular mycorrhizal fungi."

Line 130: Reword to say "Auxins are implicated in the process of AM colonization."

Author Response

Please, see the attached doccument
